# HER2 Signaling Implicated in Regulating Alveolar Epithelial Permeability with Cyclic Stretch

**DOI:** 10.3390/ijms20040948

**Published:** 2019-02-22

**Authors:** Nadir Yehya, Min Jae Song, Gladys G. Lawrence, Susan S. Margulies

**Affiliations:** 1Department of Bioengineering, University of Pennsylvania, 40 Skirkanich Hall, 210 South 33rd Street, Philadelphia, PA 19104, USA; minjae.song@nih.gov (M.J.S.); ggb@seas.upenn.edu (G.G.L.); susan.margulies@gatech.edu (S.S.M.); 2Department of Anesthesiology and Critical Care Medicine, Children’ Hospital of Philadelphia and University of Pennsylvania, Suite 7C-26, 3401 Civic Center Boulevard, Philadelphia, PA 19104, USA; 3Wallace H. Coulter Department of Biomedical Engineering, Georgia Tech College of Engineering, Emory University School of Medicine, Atlanta, GA 30332, USA

**Keywords:** ventilator-induced lung injury, VILI, HER2, HER3, neuregulin-1, NRG1

## Abstract

Mechanical ventilation can be damaging, and can cause or exacerbate ventilator-induced lung injury (VILI). The human epidermal growth factor receptor (HER) ligand neuregulin-1 (NRG1) activates HER2 heterodimerization with HER3, and has been implicated in inflammatory injuries. We hypothesized that HER2 activation contributes to VILI. We analyzed a database of differentially expressed genes between cyclically stretched and unstretched rat alveolar epithelial cells (RAEC) for HER ligands and validated the differential expression. The effect of the ligand and HER2 inhibition on RAEC permeability was tested, and in vivo relevance was assessed in a rat model of VILI. Analysis of our expression array revealed the upregulation of NRG1 and amphiregulin (AREG) with stretch. NRG1 protein, but not AREG, increased after stretch in culture media. Treatment with an NRG1-cleavage inhibitor (TAPI2) or an inhibitor of NRG1-binding (anti-HER3 antibody) reduced HER2 phosphorylation and partially mitigated stretch-induced permeability, with the upregulation of claudin-7. The results were reproduced by treatment with a direct inhibitor of HER2 phosphorylation (AG825). The transfection of microRNA miR-15b, predicted to negatively regulate NRG1, also attenuated stretch-induced permeability, and was associated with lower NRG1 mRNA levels. In rats ventilated at damaging tidal volumes, AG825 partly attenuated VILI. We concluded that cyclic stretch activates HER2 via the HER3 ligand NRG1, leading to increased permeability. Outcomes were mitigated by the downregulation of NRG1, prevention of NRG1 binding, and most strongly by the direct inhibition of HER2. In vivo HER2 inhibition also attenuated VILI. Ligand-dependent HER2 activation is a potential target for reducing VILI.

## 1. Introduction

It has been recognized since 1974 that ventilation with large tidal volumes and pressure swings can cause alveolar barrier disruption in pre-clinical models [1]. The clinical implications of ventilator-induced lung injury (VILI) were subsequently confirmed in the landmark ARDSNetwork trial demonstrating that lower tidal volumes and airway pressures in adults with acute respiratory distress syndrome (ARDS) resulted in fewer ventilator days and improved survival [2], a finding which has since been extended to patients without ARDS [3,4]. Despite the appreciation that mechanical ventilation, while life-saving, contributes to and exacerbates lung injury, there remains an incomplete understanding of the mechanism and pathophysiology underlying VILI.

Epithelial injury is a hallmark of VILI, and the loss of alveolar epithelial barrier function leads to alveolar flooding and life-threatening hypoxemia. The human epidermal growth factor receptor-2 (HER2) is expressed by pulmonary epithelial cells, and has recently been implicated in alveolar epithelial dysfunction [5,6,7]. The HER receptor family consists of four plasma membrane-bound tyrosine kinase receptors: HER1 or epidermal growth factor receptor (EGFR), HER2, HER3, and HER4 [8]. HER2 has no known ligand but can be activated by heterodimerization with any other HER receptors [9].

Neuregulin-1 (NRG1) is a HER ligand that is cleaved from pulmonary epithelial cell surfaces by a disintegrin and metalloprotease 17 (ADAM17) [10]. NRG1-mediated HER2 activation has been implicated in Interleukin-1β (IL-1β)-induced lung injury after bleomycin exposure [5]. HER3 is a receptor for NRG1, but HER3 has impaired kinase activity and requires heterodimerization with HER2, thereby activating the HER2 tyrosine kinase domain with subsequent HER2 phosphorylation and the initiation of downstream signaling [11].

Recently, our group performed genome-wide mRNA expression analysis on cyclically-stretched rat alveolar epithelial cells (RAEC) to identify candidate genes involved in stretch-induced increases in permeability [12]. Given the recent evidence implicating HER ligand-mediated HER2 activation, we queried our database for HER ligands and subsequently investigated the role of HER2 activation in VILI. We hypothesized that HER2 activation contributes to cyclic stretch-induced VILI.

## 2. Results

### 2.1. Cyclic Stretch Increases NRG1 Release

We previously performed genome-wide mRNA expression analysis in RAEC that underwent cyclic stretch at ΔSA (surface area) 12% or ΔSA 25% for 6 h, or served as unstretched controls [12]. We queried this database for known HER ligands that were differentially expressed in stretched cells (Table 1). Amphiregulin (AREG) and NRG1 were upregulated with cyclic stretch at both stretch magnitudes, which was confirmed by RT-PCR (Figure 1A). NRG1, but not AREG, increased in cell culture media after 1 h of cyclic stretch (Figure 1B). Accordingly, we focused our investigations on NRG1.

### 2.2. HER2 Activation and Epithelial Permeability

To test whether NRG1-dependent HER2 activation was involved in cyclic stretch-induced permeability, RAEC underwent cyclic stretch in the presence of TAPI2 (ADAM17 inhibitor preventing NRG1 cleavage), anti-HER3 antibody (preventing NRG1 binding to HER3), or direct inhibition of HER2 activation (AG825). Activation of HER2 was measured by Western blot analysis of phosphorylated HER2 (pHER2) after IP using anti-HER3 and (separately) anti-HER2 as bait. We reasoned that using HER3 as bait in IP would enrich for activated HER2, as ligand-dependent HER3 activation promotes heterodimerization with HER2. Cyclic stretch led to increased pHER2, which was mitigated by treatment with TAPI2, anti-HER3 antibody, and AG825 (Figure 2A). Additionally, the inhibition of ligand-dependent HER2 activation also mitigated stretch-induced increases in permeability (Figure 2B).

To test whether the improved permeability with HER2 inhibition was mediated via tight junction proteins, we assessed the expression of zonula-occludens 1 (ZO-1)-bound proteins after treatment with TAPI2 and AG825. Cyclic stretch decreased ZO-1-bound claudin-7 expression, which was prevented by treatment of both TAPI2 and AG825 (Figure 3).

### 2.3. Transfection of miR-15b

We had previously investigated and described the genome-wide differential expression of microRNA between stretched and unstretched RAEC [13]. Using TargetScan (version 6.2), we queried the differentially expressed database for miRNA predicted to target HER ligands that were anti-correlated (i.e., upregulated genes predicted by downregulated miRNA). This miR-15b, which was downregulated with cyclic stretch in our database, was predicted to target NRG1. Transfection of miR-15b in stretched RAEC resulted in lower NRG1 expression, as well as the reduction of stretch-induced increases in permeability (Figure 4), consistent with miR-15b as a promoter of epithelial barrier maintenance.

### 2.4. HER2 Inhibition Mitigates VILI In Vivo

To test whether HER2 activation has a causal role in disrupting epithelial barrier integrity in vivo, we tested the effect of pre-treatment with AG825 in a rodent model of VILI. Injurious ventilation resulted in increased alveolar permeability (Figure 5A,B) and reduced respiratory system compliance (Figure 5C), both of which were improved in the AG825 treated animals. In this model, AG825 did not affect the bronchoalveolar lavage (BAL) levels of IL-1β (Figure 5D), although BAL neutrophil activity as measured by myeloperoxidase (MPO) was reduced (Figure 5E).

## 3. Discussion

The cyclic stretch of RAEC increased NRG1 expression and release in vitro, with subsequent activation of the HER2 pathway. At the tight junction, ZO-1-bound claudin-7 was slightly reduced, with associated increases in paracellular permeability. The inhibition of NRG1 cleavage and interference with HER3 partially mitigated stretch-induced increases in permeability in vitro, whereas the direct inhibition of HER2 phosphorylation returned the permeability levels to baseline. In rats undergoing damaging ventilation, AG825 mitigated the increased permeability and compliance, suggesting the relevance of HER2 activation in VILI.

Previous studies in bleomycin-injured mice have demonstrated an essential role for NRG1 in linking IL-1β to HER2 activation, and the subsequent downstream disruption of alveolar epithelial barrier function [5]. The same group has demonstrated elevation in BAL [5,6] and plasma [6] NRG1 in adults with acute respiratory distress syndrome, suggesting that NRG1 can act as a biomarker for lung injury, and implicating NRG1-mediated HER2 activation in diverse inflammatory lung pathologies. We extend these findings to support a potential role for NRG1 and HER2 activation in in vitro and in vivo models of VILI. This has clinical relevance, as it suggests that HER2 remains an active therapeutic target even after the initial inflammatory insult, and further propagates lung injury after the initiation of mechanical ventilation. Our investigations identified four distinct potential targets for intervention: NRG1 expression, NRG1 cleavage by ADAM17, HER3 ligand binding, and HER2 phosphorylation.

Our data did not directly address whether IL-1β mediates NRG1 release, as has been shown with bleomycin-induced lung injury [5]. The cyclic stretch of epithelial cells in vitro [14] and injurious ventilation in vivo [15,16] are associated with increased IL-1β. However, it is unlikely that NRG1 release is mediated by IL-1β in our model. HER2 activation was noted with as little as 10 min of stretch and was mitigated by TAPI2 or anti-HER3 antibody treatment, suggesting rapid, ADAM17-dependent cleavage of NRG1 after stretch. The cyclic stretch of lung epithelia has only been shown to increase IL-1β after 3 h [14]. In vivo models of VILI have shown increased IL-1β after 30 to 60 min of ventilation [15,16]; however, the source of this IL-1β remains unclear. While both recruited neutrophils [15] or alveolar macrophages [17], in addition to stretched lung epithelia, have been implicated in released IL-1β during VILI, in our model, neutrophil infiltration as measured by BAL MPO was reduced with AG825, suggesting other sources of the elevated IL-1β both in animals treated and not treated with AG825. Moreover, while this additional IL-1β potentially contributes to additional NRG1 release during either VILI or any inflammatory lung insult, its impact on the increased NRG1-induced HER2 activation in our in vitro cyclic stretch system is likely negligible. In our in vivo model, the AG825-mediated improvement in permeability could potentially suggest that HER2 activation is downstream of IL-1β. However, this was not explicitly tested.

Our data suggests that NRG1 is released due to increased ADAM17 activation with cyclic stretch, rather than the mechanical disruption of NRG1 binding to cell membranes and subsequent release. Inhibition with TAPI2 reduced HER2 activation and partly mitigated stretch-induced permeability, suggesting a role for increased ADAM17 activity with cyclic stretch. TAPI2 has also been demonstrated to attenuate lung injury induced by IL-1β both in vitro and in vivo [5], consistent with a central role for ADAM17 in regulating NRG1 cleavage and HER2 activation in diverse etiologies of lung injury. Given the elevated levels of NRG1 mRNA after 6 h of stretch, however, ongoing mechanisms of NRG1 release and paracellular activation are plausible. Furthermore, TAPI2 did not attenuate stretch-induced increases in permeability as well as the direct inhibition of HER2 phosphorylation, suggesting that other pathways leading to HER2 activation, potentially through other HER ligands, may be involved in VILI.

Finigan et al. demonstrated that HER2 activation by NRG1 resulted in phosphorylation of beta-catenin, causing the subsequent dissociation of beta-catenin from E-cadherin and decreased E-cadherin-mediated cell adhesion [18]. Our results provide another potential pathway leading to increased epithelial permeability via ZO-1-bound claudin-7. However, the magnitude of worsened permeability is larger than the effect on claudin-7. This suggests that there are likely other mechanisms responsible for HER2-mediated increases in permeability.

The identification of miR-15b as a possible negative regulator of NRG1 introduces an additional avenue for VILI therapy. As we did not perform a reporter assay to definitively prove that NRG1 mRNA levels were regulated by miR-15b, the improved barrier function with miR-15b may not entirely be associated with reduced NRG1. However, the response to miR-15b and the lower NRG1 mRNA levels were consistent with a role for NRG1 in this pre-clinical model of VILI.

Our study has several limitations. We focused our investigations on NRG1 and the subsequent HER3 transactivation of HER2, as AREG was not elevated. Accordingly, other HER-family receptors, such as EGFR and HER4, were not investigated. Alternative pathways for HER2 activation exist, both involving HER ligands that were not identified in our microarray, as well as non-HER ligands. IL-6, for example, has been shown to activate HER2 by forming a complex with the IL-6 receptor. IL-6, which along with IL-1β is universally elevated after inflammatory stimuli, may also contribute to HER2 activation in vivo. Likely, multiple inflammatory stimuli converge on HER2 activation, which we demonstrate plays a central role in disrupted alveolar barrier function in experimental lung injury. In vivo inhibition of HER2 activation with AG825 in our model of injurious ventilation improved permeability, suggesting that HER2 activation was sufficiently upstream to warrant targeting in VILI and other etiologies of lung injuries. However, we did not assess whether AG825 resulted in lower levels of pHER2 in vivo, which limited our conclusions. Additionally, we could not successfully perform ZO-1 IP in cells treated with anti-HER3 antibodies, preventing an assessment of direct HER3 inhibition on claudin levels. Finally, we did not collect blood gases or histology, precluding a full assessment of the damage caused by our VILI model, and the complete nature of the protection offered by AG825.

## 4. Methods

### 4.1. Rat Alveolar Epithelial Cell Isolation

In a protocol approved by the Institutional Animal Care and Use Committee (IACUC protocol 804154, last approved 27 December 2017) of the University of Pennsylvania, alveolar type II cells were isolated from male Sprague-Dawley rats based on a method reported by Dobbs et al. [19] with a slight modification [20]. Briefly, rat lungs were isolated while continuously ventilated and perfused via the pulmonary artery to remove blood immediately following sacrifice. Lungs underwent alveolar lavage to remove debris and subsequently underwent enzymatic digestion followed by mincing. The resultant solution was depleted of immune cells by negative selection following incubation in IgG-coated plates, resulting in type II cells. Type II cells were seeded onto fibronectin coated (10 μg/cm^2^) flexible silastic membranes (Specialty Manufacturing, Saginaw, MI, USA), and mounted in custom designed wells at a density of 10^6^ cells/cm^2^. Cells were cultured for 5 days with MEM supplemented with 10% fetal bovine serum (FBS), until they attained alveolar type I cell characteristics [12,13,21].

### 4.2. Cyclic Stretch

Type I-like RAEC were serum-deprived with 20 mM Hepes supplemented with DMEM (CO_2_ free buffering system) for 2 h, and subjected to biaxial cyclic stretch (∆SA 25% or ΔSA 37%) at 37 °C at a frequency of 0.25 Hz (15 cycles/min). Cells were stretched for 10 min (phosphorylation experiments), 1 h (HER ligand secretion experiments), or 6 h (gene expression and miRNA transfection experiments). In some experiments, cells were treated for 2 h prior to and during stretch with either the ADAM17 inhibitor TAPI2 (50 µM; Calbiochem, Burlington, MA, USA), competitive anti-HER3 antibody to prevent ligand binding (10 µg antibody/mL DMEM; Abcam, Cambridge, MA, USA), the HER2 tyrosine kinase inhibitor AG825 (50 µM; Calbiochem), or vehicle control (DMSO).

### 4.3. RT-PCR

Total RNA was extracted from the cells (Qiagen miRNAeasy mini kit cat# 217004, Qiagen Inc, Valencia, CA, USA) per manufacturer’s instructions. The quantity and quality of the RNA samples was measured (Agilent Bioanalyzer and Nanodrop spectrophotometer, Santa Clara, CA, USA). First strand cDNA synthesis occurred in 20 µL reactions with sequence specific Taqman primers, per manufacturer’s protocol. PCR was performed using Taqman universal master mix. Rat 4.5s RNA was used as an endogenous control for miRNA RT-PCR, and GAPDH used for mRNA. Values were normalized to endogenous controls using the Δ*C*_t_ method, and subsequently to the unstretched control group.

### 4.4. Western Blotting of Secreted HER Ligands

After 1 h of cyclic stretch at ΔSA 37% (or unstretched control condition), RAEC media were collected, protein concentrated, and quantified by Bradford assay. Equal amounts (60 µg) of the protein was run on SDS-polyacrylamide (4–12%) gels, transferred onto polyvinylidene fluoride membranes (PVDF), and non-specific binding was blocked in TBS containing 5% non-fat powdered milk and 0.1% Tween-20 at room temperature. Membranes were probed for AREG or NRG1 (both antibodies from Santa Cruz Biotechnology, Dallas, TX, USA). Specific activity was calculated through densitometric analysis (Kodak, Rochester, NY, USA). Band intensities were normalized to unstretched control cells.

### 4.5. Co-Immunoprecipitation and Western Blotting

After 10 min of cyclic stretch at ΔSA 37% (or unstretched control condition), RAEC were washed with PBS and scraped from silastic membranes. To measure active phosphorylated HER2 (pHER2), we used the Pierce Classic Immunoprecipitation (IP) Kit to enrich for pHER2. Given that NRG1 binds HER3 and induces heterodimerization with HER2, we performed IP (50 µg bait antibody) using (separately) either anti-HER3 or anti-HER2 (both antibodies from Cell Signaling Technologies, Danvers, MA, USA). After elution of immune complex, 60 µg of the protein lysate was run on SDS-polyacrylamide (4–12%) gels, transferred onto PVDF, and non-specific binding blocked in TBS containing 5% non-fat powdered milk and 0.1% Tween-20 at room temperature. Membranes were probed for pHER2 (phospho-Tyr1248, Cell Signaling Technologies).

A similar method was used to measure tight junction proteins binding to ZO-1. IP was performed using anti-ZO-1 (Cell Signaling Technologies), and Western blotting performed as above probing for occludin (Zymed/Invitrogen, Waltham, MA, USA), claudin-4, claudin-7, or claudin-18 (all claudin antibodies from Zymed/Invitrogen). Band intensities for probed proteins were normalized to unstretched controls.

### 4.6. MicroRNA Transfection

Using above described methods, we isolated RAEC and cultured it for 3 days in antibiotic free MEM plus 10% FBS. On day 4 of culture, we transfected a custom-designed miR-15b mimic, or a scrambled negative control (Exiquon, Vedbaek, Denmark). These constructs use a locked nucleic acid (LNA) technology, and were conjugated to a Texas Red dye for visual confirmation of transfection efficiency. RAEC were transfected with miR-15b or negative control using Lipofectamine 2000 (Invitrogen) reagent diluted in Opti-MEM for a final miRNA concentration of 80 nM, per manufacturer’s protocol. Optimum transfection concentration was determined by titration of different concentrations of inhibitors (50 to 100 nM) and evaluating monolayer fluorescence after 24 h, using >90% cellular fluorescence as a threshold for efficient transfection. On day 5, RAEC were serum-deprived with 20 mM Hepes supplemented with DMEM for 2 h, subjected to biaxial cyclic stretch (∆SA 25%) at 37 °C for 6 hat a frequency of 0.25 Hz (15 cycles/min), or served as unstretched controls.

### 4.7. Cell Permeability Assays

To test RAEC permeability, we used ouabain, a small uncharged molecule, with high affinity for basolateral Na^+^/K^+^-ATPase, which when conjugated to the nonpolar fluorophore BODIPY, served as a tracer. Our group has validated this method before [13,22,23], and demonstrated that BODIPY-ouabain binding represents loss of tight junction integrity and increased paracellular permeability, and that increased fluorescent signal is not a result of cell death or intracellular uptake or transport. BODIPY-ouabain (Invitrogen) diluted in dimethyl sulfoxide was added to cells at a final concentration of 2 µM, and incubated for 1 h. At the end of the hour, all cells were washed with DMEM to remove excess BODIPY-ouabain from the apical surface, and were visualized under a fluorescent microscope using a green emission filter. Presence of fluorescence was evidence of high affinity ouabain binding to the basolateral surface, and used as a measure of paracellular permeability. Fluorescence was measured on 4 separate fields per well, and 3 wells were measured per treatment group per animal.

### 4.8. In Vivo Model of Ventilator-Induced Lung Injury

This protocol was approved by the University of Pennsylvania IACUC. Male Sprague-Dawley rats (aged 8–10 weeks) were randomized to receive either vehicle (DMSO) or AG825 (1.67 mg/kg/day) intraperitoneally (0.5 mL daily) for 3 total doses (total 5 mg/kg). On day 3, rats were anesthetized with pentobarbital (50 mg/kg intraperitoneal), underwent tracheostomy with a stiff 14G catheter, and received a subcutaneous 10 mL saline bolus. FITC-labeled albumin (Sigma Aldrich) was intravenously injected (0.5 mL of 15 mg/mL solution diluted in PBS) prior to ventilation. Rats underwent injurious ventilation for 4 h (V_T_ 25 mL/kg, end-expiratory pressure 0 cmH_2_O, rate 27 breaths/minute, FIO_2_ 0.21) using a small-animal ventilator (Harvard Apparatus, Holliston, MA, USA). Non-ventilated, vehicle-treated rats served as a comparison group. Respiratory system static compliance was assessed by attaching the tracheal cannula to a pressure manometer/capnograph (Harvard Apparatus), and plateau pressure recorded at the end of 2 h of ventilation during an inspiratory hold during volume control ventilation, with tidal volume recorded within the ventilator. Non-ventilated control rats underwent tracheal cannulation and attachment to a pressure manometer and ventilator for measurement of compliance similar to ventilated rats.

At the end of the experiment, blood was collected via ventricular puncture into citrated tubes, and rats sacrificed by exsanguination. Citrated blood was centrifuged (2000 g, 20 min, 20 °C) to generate platelet-poor plasma. Rats underwent BAL using 24 mL of PBS (Ca^2+^ and Mg^2+^ added) in 3 separate aliquots of 8 mL each. Each aliquot was centrifuged (300 g, 10 min, 10 °C), and the cell-free supernatant recovered. Fluorescence was measured in both the BAL supernatant and the plasma, and permeability was assessed by the ratio of BAL: blood fluorescence. BAL total protein levels were measured by the DC Protein Assay (BioRad, Hercules, CA, USA). BAL levels of IL-1β were measured using Rat IL-1β ELISA (RayBiotech Inc., Norcross, GA, USA). BAL neutrophil activity was quantified using a myeloperoxidase (MPO) ELISA (R & D Systems, Minneapolis, MN, USA).

### 4.9. Statistical Analysis

Values are expressed as means ± SEM. Parametric statistics, including unpaired *t*-tests or ANOVA were used as appropriate. ANOVA was followed by post-hoc Tukey test. *p* < 0.05 was considered statistically significant. To standardize reporting of different measurements in RAEC, (RNA by RT-PCR, protein intensity by Western, fluorescence intensity by imaging), values are normalized to unstretched control cells.

## 5. Conclusions

We demonstrated that the cyclic stretch of alveolar epithelia produced HER2 activation by the HER3 ligand NRG1, which led to increased permeability in vitro. The downregulation of NRG1 by TAPI2 or miR-15b, prevention of NRG1 binding by an anti-HER3-antibody, or direct inhibition of HER2 activation each improved epithelial barrier function. In vivo inhibition of HER2 activation with AG825 pre-treatment also attenuated VILI. HER2 activation is a potential target for reducing VILI.

## Figures and Tables

**Figure 1 ijms-20-00948-f001:**
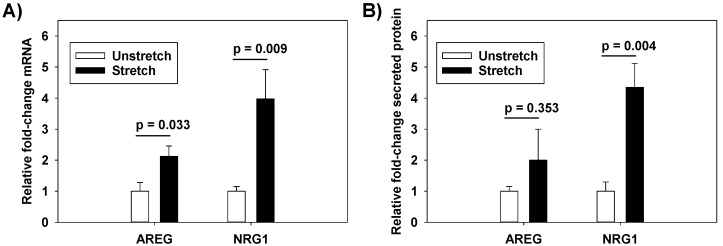
Increases in (**A**) mRNA and (**B**) secreted protein with cyclic stretch (ΔSA 37%, 0.25 Hz, 1 h) as measured by RT-PCR or Western blot analysis. Values are normalized to unstretched cells. *n* = 5; *p* values represent the result of unpaired *t*-tests.

**Figure 2 ijms-20-00948-f002:**
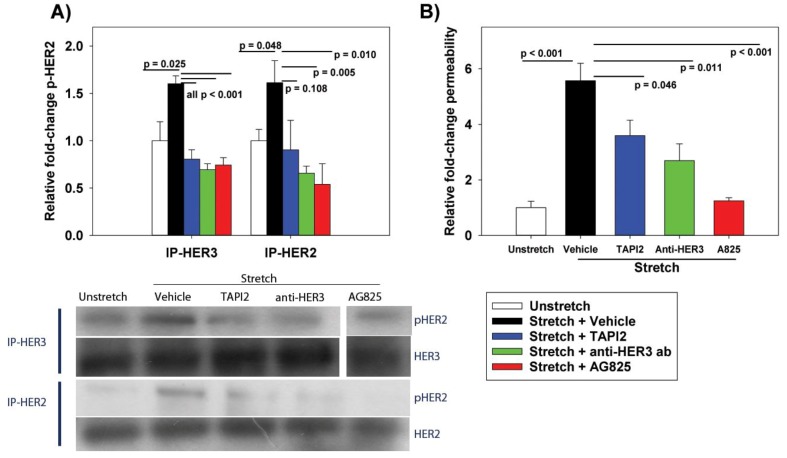
Effect of cyclic stretch (ΔSA 37%, 0.25 Hz, 10 min) and HER pathway inhibition on (**A**) human epidermal growth factor receptor-2 (HER2) activation and (**B**) permeability. (**A**) HER2 activation was measured by Western blot analysis of pHER2 after IP by (separately) anti-HER3 or anti-HER2 antibody. (**B**) Permeability was measured by BODIPY-ouabain fluorescence. Treatment with the NRG1-cleavage inhibitor TAPI2 (50 µM), anti-HER3 antibody (10 µg/mL), and HER2 phosphorylation AG825 (50 µM) mitigated stretch-induced increases in pHER2 and permeability. Values are normalized to unstretched cells. *n* = 5; *p* values represent the result of ANOVA and post-hoc Tukey.

**Figure 3 ijms-20-00948-f003:**
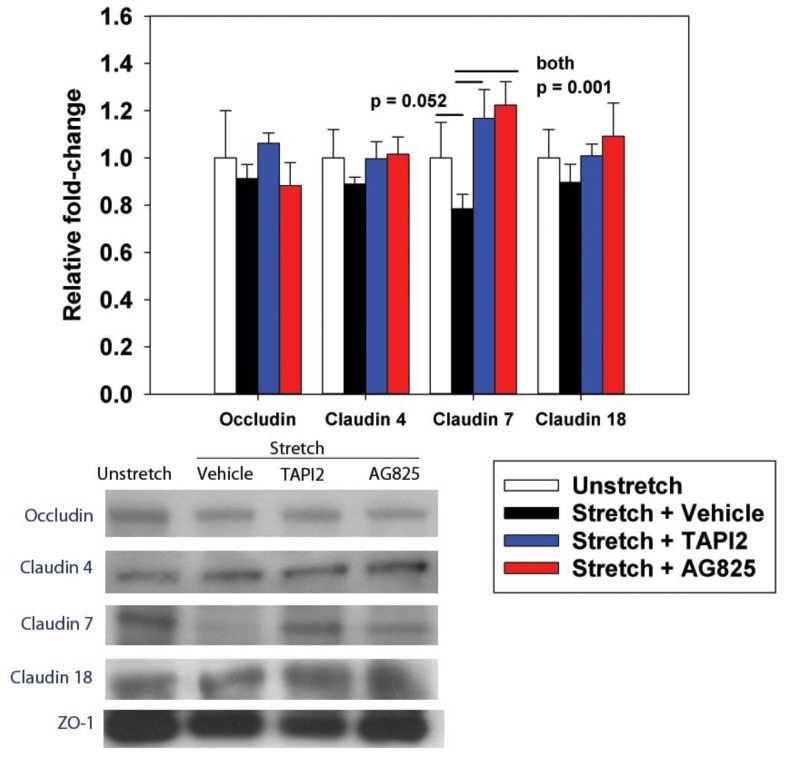
Effect of cyclic stretch (ΔSA 37%, 0.25 Hz, 10 min) and HER pathway inhibition on tight junction protein expression. After IP using ZO-1, proteins were quantified using Western blot analysis for occluding, and claudins-4, 7, and 18. Treatment with TAPI2 (50 µM) and AG825 (50 µM) reversed stretch-induced decreases in claudin-7. Values are normalized to unstretched cells. *n* = 5; *p* values represent the result of ANOVA and post-hoc Tukey.

**Figure 4 ijms-20-00948-f004:**
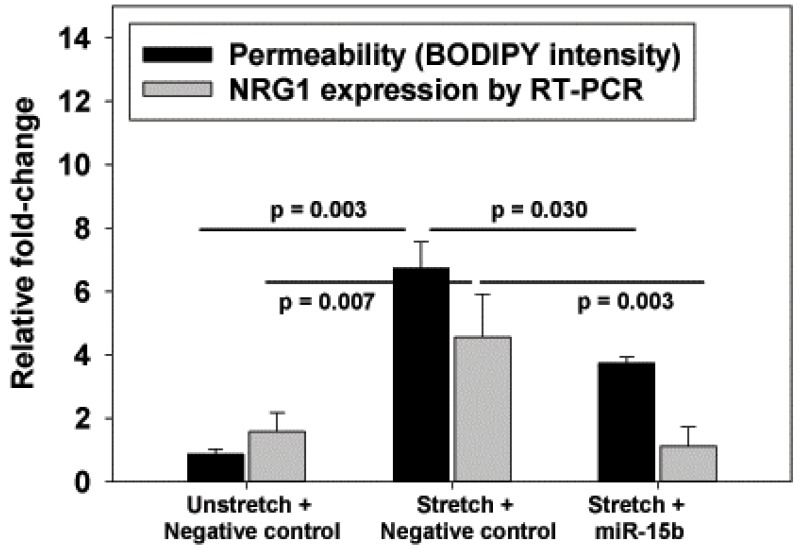
Effect of exogenous treatment with miR-15b (80 nM) or scrambled negative control on rat alveolar epithelial cells (RAEC) subject to cyclic stretch (ΔSA 25%, 0.25 Hz, 6 h). MiR-15b reduced stretch-induced increases in permeability and NRG1 mRNA levels. Values are normalized to unstretched cells. *n* = 5; *p* values represent the result of ANOVA and post-hoc Tukey (performed separately for permeability and NRG1 mRNA levels).

**Figure 5 ijms-20-00948-f005:**
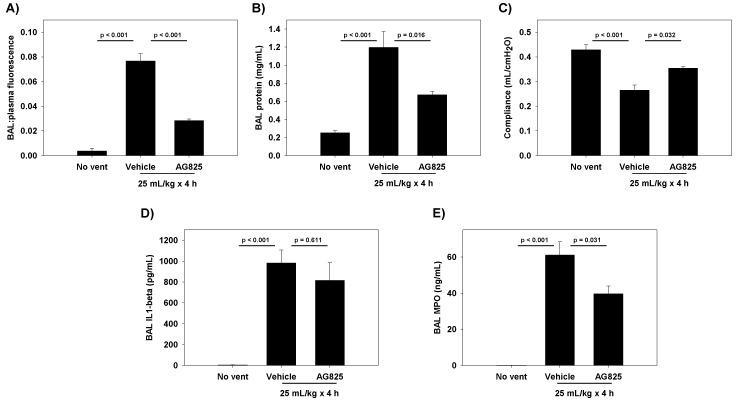
Effect of pre-treatment with AG825 (1.67 mg/kg IP × 3 days) on (**A**) lung permeability as measured by bronchoalveolar lavage (BAL): plasma fluorescence, (**B**) BAL protein concentration, (**C**) respiratory system compliance, (**D**) BAL Interleukin-1β, and (**E**) BAL myeloperoxidase levels in rats subject to injurious ventilation for 4 h (V_T_ 25 mL/kg, end-expiratory pressure 0 cmH_2_O, rate 27 breaths/min, FIO_2_ 0.21). AG825 improved permeability and compliance in this ventilator-induced lung injury (VILI) model, relative to vehicle (DMSO). *n* = 4; *p* values represent the result of ANOVA and post-hoc Tukey.

**Table 1 ijms-20-00948-t001:** Human epidermal growth factor receptor (HER) ligands upregulated with 6 h of cyclic stretch of rat alveolar epithelial cells.

mRNA	ΔSurface Area 12%	ΔSurface Area 25%
Amphiregulin (AREG)	4.3-fold	3.8-fold
Neuregulin-1 (NRG1)	3.6-fold	3.6-fold

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
