# Peer review of "HER2 Signaling Implicated in Regulating Alveolar Epithelial Permeability with Cyclic Stretch"

_ijms, 2019, doi:10.3390/ijms20040948_

Reviewer 1 Report

I cannot decide on the quality /potential impact of the presented study without receiving additional evidences and/or proper answers on relevant issues regarding the previous version of the manuscript.

It is clear that each study has its scope and limitations. However, However, addressing some important open questions as limitations of the study is not a proper way to improve a manuscript.

 Q1. There is no former prove that the AG825 worked in vivo. The authors should present data showing AG825 activity (reduced HER2 phosphorylation).

A1 “We agree that this would be ideal. However, we lack the expertise to perform Westerns

in whole lung tissue, and BAL samples did not detect HER2. Thus, we acknowledge this

as a limitation in Discussion.”

 -You have expertise to perform Western blot of phosphor IP preparations using baits to enrich HERP2 (not mentioning all other sophisticated techniques used in the study) but you are not skilled enough to perform a WB from a tissue homogenate? I find this particular answer kind of ridiculous.

 Q4. The only indicators of lung injury in the present study are relative changes in the respiratory compliance and lung permeability (the in vivo results suggest compromised endothelial cells barrier too?). Lung injury should be supported by histology, respiratory parameters (pO2, pCO2,sO2) and inflammation markers (at least IL-1b)!

 A4. We regret that we did not perform histology and blood gases. We have, however, added

BAL IL-1b to our results. We have acknowledged this as a limitation at the end of the

Discussion section.

 -Showing similar and sky-high Il-1b levels in both ventilated groups demonstrate the need of formal prove of ventilator-induced/ AG825-reduced lung injury. Also to support your answer to Q5!); And to support your speculation about the infiltrate dependent increase of pro-inflammatory cytokine.  Further, you should have noticed that I am asking for histology or WB not for W/D ratio or infiltrate count in BALF.

 Q5. Compliance of (presumably) anaesthetized animals as control is questionable. Why authors did not used protectively ventilated group as a control?

 A5. While we agree that the addition of a lung-protective ventilation group would have added some insight, we felt that since the major hypothesis being tested was whether AG825 limited lung injury caused by injurious ventilation, the addition of a lung-protective group would offer little added utility above the non-ventilated controls.

 -The question rather concerns compliance comparison between anaesthetized and spontaneous breathing and injurious ventilated animals.  

 Q6. Please provide proper description on how the compliance in the “no vent” group was measured/calculated.

 A6. Non-ventilated rats were briefly attached to a ventilator and manometer for measurement of compliance. The Methods section has been modified to now include:

“Non-ventilated control rats underwent tracheal cannulation and attachment to a pressure manometer and ventilator for measurement of compliance similar to ventilated rats.”

 - With inspiratory hold you calculate/measure static compliance whereas compliance measured during ventilation might be a dynamic one. Static and dynamic compliance are not suitable for direct comparison.. (this section is still not precisely described). Pressure is not enough for compliance calculation. It is not clear how did you measure the volume of the inhaled air in the spontaneous breading animals (pneumotachograph build in vent. device?)  

 6a. What was the rational to include the weight in the compliance calculation ml/kg*cmH2O? Presumably the animal’s weight was considered in the calculation of the tidal volume?

 A6a. While rodent weights were similar (between 280 and 340 grams), and while somemanuscripts report compliance as mL/cmH2O), we (Yehya et al, AJP-Lung 2015) and others have reported compliance normalized to weight (ml/kg/cmH2O). Anecdotally,

this is more common among pediatric intensive care practitioners, as children frequently

have physiologic values normalized to either body weight or body surface area.

 -That you and others have previously normalized compliance to body weight is not a rational answer.  Why normalizing animals with similar weight (at all)?  

Minor points

 Abstract „inhomogeneous lung inflation“ is unlikely direct or only cause of VILI

Results Figure 1. “Values are normalized to unstretched cells” in panel A) NRG1 control is not =1.

Sometimes I like a few preceding words within the results section. In the presented paper however results consist lots of introduction/discussion. As a reader I don’t like result description in the discussion section…You may consider improving paper style.

Author Response

I cannot decide on the quality /potential impact of the presented study without receiving additional evidences and/or proper answers on relevant issues regarding the previous version of the manuscript.  It is clear that each study has its scope and limitations. However, However, addressing some important open questions as limitations of the study is not a proper way to improve a manuscript.

We respect the Reviewer's position.  However, several of his concerns simply cannot be answered by our group.

 Q1. There is no former prove that the AG825 worked in vivo. The authors should present data showing AG825 activity (reduced HER2 phosphorylation).

 A1 “We agree that this would be ideal. However, we lack the expertise to perform Westerns

in whole lung tissue, and BAL samples did not detect HER2. Thus, we acknowledge this

as a limitation in Discussion.”

 -You have expertise to perform Western blot of phosphor IP preparations using baits to enrich HERP2 (not mentioning all other sophisticated techniques used in the study) but you are not skilled enough to perform a WB from a tissue homogenate? I find this particular answer kind of ridiculous.

We regret that the Reviewer did not find this answer satisfactory.  Nevertheless, despite its apparent ridiculousness, the answer is the truth.  Westerns from cell cultures are markedly more simple, and are an established procedure in our group.  We have only recently extrapolated to animals, and our attempts at tissue homogenation and subsequent Westerns have not been successful.  Thus, our answer stands.

Q4. The only indicators of lung injury in the present study are relative changes in the respiratory compliance and lung permeability (the in vivo results suggest compromised endothelial cells barrier too?). Lung injury should be supported by histology, respiratory parameters (pO2, pCO2,sO2) and inflammation markers (at least IL-1b)!

A4. We regret that we did not perform histology and blood gases. We have, however, added

BAL IL-1b to our results. We have acknowledged this as a limitation at the end of the

Discussion section.

-Showing similar and sky-high Il-1b levels in both ventilated groups demonstrate the need of formal prove of ventilator-induced/ AG825-reduced lung injury. Also to support your answer to Q5!); And to support your speculation about the infiltrate dependent increase of pro-inflammatory cytokine.  Further, you should have noticed that I am asking for histology or WB not for W/D ratio or infiltrate count in BALF.

We do  not have residual lung tissue; only residual BAL.  Thus, we are unable to provide the requested histology or respiratory parameters.  The provided data gives a measure of permeability (fluorescence), an in vivo metric of physiology (compliance), and a metric of inflammation (BAL IL-1beta).  While incomplete, these data do point to an injury caused by aggressive ventilator settings, and mitigation by AG825.

 Q5. Compliance of (presumably) anaesthetized animals as control is questionable. Why authors did not used protectively ventilated group as a control?

A5. While we agree that the addition of a lung-protective ventilation group would have added some insight, we felt that since the major hypothesis being tested was whether AG825 limited lung injury caused by injurious ventilation, the addition of a lung-protective group would offer little added utility above the non-ventilated controls.

-The question rather concerns compliance comparison between anaesthetized and spontaneous breathing and injurious ventilated animals. 

IThe question we are addressing is whether AG825 mitigated damage in VILI.  The spontaneously breathing group simply serves as an external reference.  Thus, there is no utility, in our view, of the group suggested by the Reviewer, as it has no relevance to our question.

 Q6. Please provide proper description on how the compliance in the “no vent” group was measured/calculated.

A6. Non-ventilated rats were briefly attached to a ventilator and manometer for measurement of compliance. The Methods section has been modified to now include:

“Non-ventilated control rats underwent tracheal cannulation and attachment to a pressure manometer and ventilator for measurement of compliance similar to ventilated rats.”

- With inspiratory hold you calculate/measure static compliance whereas compliance measured during ventilation might be a dynamic one. Static and dynamic compliance are not suitable for direct comparison.. (this section is still not precisely described). Pressure is not enough for compliance calculation. It is not clear how did you measure the volume of the inhaled air in the spontaneous breading animals (pneumotachograph build in vent. device?)  

 We agree with the Reviewer that static and dynamic compliance are to be differentiated.  Furthermore, we regret not being as detailed as requested.  This section has been updated in Methods to now read:

"Respiratory system static compliance was assessed by attaching the tracheal cannula to a pressure manometer/capnograph (Harvard Apparatus), and plateau pressure recorded at the end of 2 hours of ventilation during an inspiratory hold during volume control ventilation, with tidal volume recorded within the ventilator."

6a. What was the rational to include the weight in the compliance calculation ml/kg*cmH2O? Presumably the animal’s weight was considered in the calculation of the tidal volume?

A6a. While rodent weights were similar (between 280 and 340 grams), and while somemanuscripts report compliance as mL/cmH2O), we (Yehya et al, AJP-Lung 2015) and others have reported compliance normalized to weight (ml/kg/cmH2O). Anecdotally,

this is more common among pediatric intensive care practitioners, as children frequently

have physiologic values normalized to either body weight or body surface area.

-That you and others have previously normalized compliance to body weight is not a rational answer.  Why normalizing animals with similar weight (at all)? 

We report it this way because this is an accepted and standard way to report compliance.

Frey et al.  AJP-Lung 2010

Jeng et al. Pediatric Research 2009

Chen et al. Journal of Thoracic Disease 2014

Minor points

 Abstract „inhomogeneous lung inflation“ is unlikely direct or only cause of VILI

We have changed this in Abstract to now read:

"Mechanical ventilation can be damaging, and can cause or exacerbate ventilator-induced lung injury (VILI)"

 Results Figure 1. “Values are normalized to unstretched cells” in panel A) NRG1 control is not =1.

The Reviewer is correct.  This was mis-transcribed in our graphing program, resulting in the erroneous figure.  This has been corrected.

Sometimes I like a few preceding words within the results section. In the presented paper however results consist lots of introduction/discussion. As a reader I don’t like result description in the discussion section…You may consider improving paper style.

We have attempted to remove as much of the Results descriptions in the Discussion.

Reviewer 2 Report

Authors have addressed most of my concerns, although they cannot provide better blots. And the manuscript has improved.

Author Response

We appreciate the Reviewer's time and effort.

Round  2

Reviewer 1 Report

I appreciate the author’s efforts to improve the manuscript.

-Although suggested by the in vitro data, a direct prove of AG825 activity in vivo is not shown because of missing expertises. 

I appreciate the frank answer. It however cannot fill this data gap. Further it makes me wonder how the animal experiments did receive a permit... (Just a comment … don’t need to discuss)

- According to Matute-Bello 2011 AmJRespCellMol Biol vol 44, pp7725-738 one need 3 from 4 “Features and measurements” in respect to designate an animal model an ALI-model.  In the present paper only two parameters (Permeability and Inflammation) are shown and they appear contradictory in respect to inhibited (but not shown) HER2 phosphorylation. The suggestion/speculation:  “In our in vivo model, the AG825-mediated improvement in permeability suggests that HER2 activation is downstream of IL-1β” (P6L161-3) needs a prove or at least relevant discussion with literature support.

In the particular case the missing histology definitely impairs data interpretation/understanding.

A possible way to partially solve this issue might be to provide a measure of neutrophil count/activity (e.g. MPO in BALF) -It will be difficult to believe that it is possible to maintain alveolar barrier in the presence of active neutrophils. It is suggested by the authors that IL-1b originates from the inflammatory neutrophils P6L156-7 [15]. On die other hand tight alveolar barrier should be related to low levels of inflammatory infiltrates?

Further total protein/IgM/…  in BALF as alternative method to illustrate edema  (also to exclude an artefact from the FITC-albumin method (e.g. linear relation between concentration and fluorescent signal?)).  

-I would like to see the rational not a list of papers showing compliance /kg (the list is long I don’t doubt it). So I will repeat the question:

Taking into account that “rodent weights were similar“, what was the rational to include the weight in the compliance calculation?

I am curious to see the compliance data without normalization for weight? And furthermore to see in which manner the FITC-albumin correlates to lung compliance (with and without normalization)? The authors should provide the related raw data in a table form for reviewer consideration.     

Author Response

I appreciate the author’s efforts to improve the manuscript.

-Although suggested by the in vitro data, a direct prove of AG825 activity in vivo is not shown because of missing expertises. 

I appreciate the frank answer. It however cannot fill this data gap. Further it makes me wonder how the animal experiments did receive a permit... (Just a comment … don’t need to discuss)

- According to Matute-Bello 2011 AmJRespCellMol Biol vol 44, pp7725-738 one need 3 from 4 “Features and measurements” in respect to designate an animal model an ALI-model.  In the present paper only two parameters (Permeability and Inflammation) are shown and they appear contradictory in respect to inhibited (but not shown) HER2 phosphorylation. The suggestion/speculation:  “In our in vivo model, the AG825-mediated improvement in permeability suggests that HER2 activation is downstream of IL-1β” (P6L161-3) needs a prove or at least relevant discussion with literature support.

In the particular case the missing histology definitely impairs data interpretation/understanding.

A possible way to partially solve this issue might be to provide a measure of neutrophil count/activity (e.g. MPO in BALF) -It will be difficult to believe that it is possible to maintain alveolar barrier in the presence of active neutrophils. It is suggested by the authors that IL-1b originates from the inflammatory neutrophils P6L156-7 [15]. On die other hand tight alveolar barrier should be related to low levels of inflammatory infiltrates?

We appreciate the Reviewer's suggestion.  Indeed, we feel this additional data has significantly improved the manuscript and strengthened our conclusions.  We have performed an MPO ELISA on the remaining BAL fluid, and have added the results to Figure 5 (new Figure 5).  We have updated the Results and Discussion section to accommodate these new findings.  Additionally, we have revised our previous assertion, and now state this in Discussion:

"In our in vivo model, the AG825-mediated improvement in permeability potentially could suggest that HER2 activation is downstream of IL-1β.  However, this was not explicitly tested."

We have also added text in the Discussion about the improvements in compliance and permeability, but not IL-1beta, and mention the uncertainty regarding where the IL-1beta is coming from.

Further total protein/IgM/…  in BALF as alternative method to illustrate edema  (also to exclude an artefact from the FITC-albumin method (e.g. linear relation between concentration and fluorescent signal?)).  

We have also added total protein from BAL fluid to our results (new Figure 5).

-I would like to see the rational not a list of papers showing compliance /kg (the list is long I don’t doubt it). So I will repeat the question:

Taking into account that “rodent weights were similar“, what was the rational to include the weight in the compliance calculation?

I am curious to see the compliance data without normalization for weight? And furthermore to see in which manner the FITC-albumin correlates to lung compliance (with and without normalization)? The authors should provide the related raw data in a table form for reviewer consideration.   

We think we (finally) understand the Reviewer's concern.  To be clear, as we actually SET the tidal volumes to = 25 mL/kg, we simply carried this forward when calculating compliance.  We have now changed this figure (new Figure 5) to reflect simply compliance (mL/cmH2O).  We have also attached a table for him to see the raw data.

Round  3

Reviewer 1 Report

Authors provided additional experiments and revised correspondingly the manuscript.

It is now released from preventable and/or not directly supported from the presented results conclusions and statements. Remaining speculations are clearly denoted.

I can finally suggest that the manuscript is now suited for the IJMS readers and could be published in its present form.

This manuscript is a resubmission of an earlier submission. The following is a list of the peer review reports and author responses from that submission.

Round  1

Reviewer 1 Report

In this manuscript, the authors use primary alveolar epithelial cells as well as animal modeling to demonstrate that HER2 ligand, NRG1 promoted stretch-induced permeability, while inhibition of HER2 attenuated ventilator-induced lung injury (VILI).

These experiments were well designed, controlled, and appropriate methodology was used.  Data is presented to support this overall conclusion and the use of rat modeling is convincing.  In general, the authors provide a novel insight into the role of HER2 signaling pathway during VILI.

Major points:
1.    I would like to see more information concerning the methods used as opposed to digging through previous manuscripts to find the details.  Besides, more details should be included in the figure legends.  For example, the dose of inhibitors and time points in all figure.

2.    It is hard to understand why authors included miR-15a in the manuscript. Moreover, the data in Figure 4 is not enough to support the conclusion that miR-15a target NRG1.  

3.    Western data showing in Figure 2. The band is either too dark or too weak. Please provide a better result. Similar concern in Figure 3.

Reviewer 2 Report

I cannot suggest the manuscript for publication in its present form

 AG825 inhibitor and miR-15b (in vitro) showed (separately) protective potential against VILI and stretch-induced epithelial permeability.

However, the major statement about the role of HER2 in epithelia cell permeability is not supported by the data presented in the manuscript as pointed below.

 Some controls are missing or not discussed. The discussion is patchy includes overstatements and somehow contradictory/redundant therefore unconvincing.

 Major concern

I don’t find sufficient proves on the role of HER2 in epithelial permeability in the presented manuscript.

 In vivo

1. There is no former prove that the AG825 worked in vivo.  The authors should present data showing AG825 activity (reduced HER2 phosphorylation).

2. The in vivo setup did not consider possible effect of the vehicle (DMSO) on permeability and lung compliance. Further, control non-ventilated +AG825 may hint to intrinsic effects of the inhibitor.

3. The IL-1b might by “negligible” (L159) for the short in vitro experiments. I would expect however that pro-inflammatory IL levels (including IL-1b; IL-6) will be high enough after 2 h mechanical ventilation with TV of 25ml/kg and PEEP 0 (as discussed indeed). So IL-1b may well act in the in vivo setup.  Please, provide IL-1b levels in BALF (and or in serum/plasma).

4. The only indicators of lung injury in the present study are relative changes in the respiratory compliance and lung permeability (the in vivo results suggest compromised endothelial cells barrier too?). Lung injury should be supported by histology, respiratory parameters (pO2, pCO2, sO2) and inflammation markers (at least IL-1b)!

5. Compliance of (presumably) anaesthetized animals as control is questionable. Why authors did not used protectively ventilated group as a control?

6. Please provide proper description on how the compliance in the “no vent” group was measured/calculated.

6a. What was the rational to include the weight in the compliance calculation ml/kg*cmH2O? Presumably the animal’s weight was considered in the calculation of the tidal volume?

 In vitro

7. All three blocking approaches TAPI2, anti-Her3 and AG825 reduce HER2 phosphorylation (Fig.2)! In addition, bothTAPI2 and AG825 (antiHer3Ab group was omitted in this setup?) preserved claudin-7 levels ((unlikely expression as mentioned in the text (L96) since data on expression or new synthesis are not shown)) (Fig3)!! However, permeability remains high (I assume significant although not indicated??) for TAPI2 and Anti Her3 treated cells (Fig.2) despite reduced HER2 phosphorylation. These results did not support the major statement about the proposed role of HER2 and deserved explanation/discussion.

8. L178 “pleiotropy and ability to regulate multiple related pathways with a single intervention” is an advantage of potential miRNA therapy. Pure overstatement! Without supporting data/studies is such a statement merely speculative. 8a. However I completely agree with the authors that the experiment carried showed that “the improved barrier function with miR-15b in stretched cells may not entirely be associated with reduced NRG1” (L179-180).

9. The data presented in the manuscript on the role of ZO-1-baound claudin 7 are not sufficient to support the proposed role in maintenance of epithelia permeability. The related discussion (L168-) sounds redundant.

10. L131 “Cyclic stretch of RAEC increased NRG1 expression and release in vitro, with subsequent activation of the HER2 pathway.” If presented correctly in your introduction NRG1 is not simply released but rather actively processed by ADAM17. Is the proposed pathway possible without NRG1 cleavage? ADAM17 plays an important role in the proposed mechanism.  As actually discussed later (L160-) (un example for not well structured discussion). In this case it is odd that ADAM17 levels/activity were not considered (in vitro and in vivo).

11. Did the authors showed significance only vs. stretch+vehicle group? If yes what is the rational?

 Minor points

It is not necessary to described previous work in the abstract (L17-). This explanation appeared in the introduction (L57-) and results (L66-; 105-;) sections.

L51-52 “NRG1 is a HER ligand which is cleaved from pulmonary epithelial cell surfaces by a disintegrin and ADAM17. L78-79 “ADAM17 inhibitor preventing NRG1 release” NRG1 release or cleavage?

L57-58 “Recently, our group performed genome-wide mRNA expression analysis on cyclically-stretched

rat alveolar epithelial cells (RAEC) to identify candidate genes involved in VILI” . To my opinion in such model you can identify only genes related to the mechanical strain/load. VILI is much more complex to be addressed to such “simple” in vitro model.  

L122 “Injurious ventilation resulted in worse alveolar permeability” the authors may consider the use of increased instead

L135 “HER2 inhibition” do you mean inhibition of HER2 phosphorylation?

Fig.2 Why inserts for the bands of pHER2 and HER3 in the representative WB (AG825 group, IP-HER3 run)?